# How Many Private Data Are Needed for Deep Learning in Lung Nodule Detection on CT Scans? A Retrospective Multicenter Study

**DOI:** 10.3390/cancers14133174

**Published:** 2022-06-28

**Authors:** Jeong Woo Son, Ji Young Hong, Yoon Kim, Woo Jin Kim, Dae-Yong Shin, Hyun-Soo Choi, So Hyeon Bak, Kyoung Min Moon

**Affiliations:** 1ZIOVISION, Chuncheon 24341, Korea; jeongwoo12@ziovision.co.kr (J.W.S.); yooni@kangwon.ac.kr (Y.K.); 2Division of Pulmonary and Critical Care Medicine, Department of Medicine, Chuncheon Sacred Heart Hospital, Hallym University Medical Center, Chuncheon 24253, Korea; mdhong@hallym.or.kr; 3Department of Computer Science and Engineering, College of IT, Kangwon National University, Chuncheon 24341, Korea; 4Department of Internal Medicine, Kangwon National Universtiy, Chuncheon 24341, Korea; pulmo2@kangwon.ac.kr; 5KNU-Industry Cooperation Foundation, Kangwon National Universtiy, Chuncheon 24341, Korea; dyshin@kangwon.ac.kr; 6Department of Radiology and Research Institute of Radiology, University of Ulsan College of Medicine, Asan Medical Center, Seoul 05505, Korea; 7Department of Pulmonary, Allergy and Critical Care Medicine, Gangneung Asan Hospital, University of Ulsan College of Medicine, Gangneung 25440, Korea

**Keywords:** lung nodule, radiologist, deep learning, computed tomography, nodule detection, publicly available data, transfer learning

## Abstract

**Simple Summary:**

The early detection of lung nodules is important for patient treatment and follow-up. Many researchers are investigating deep-learning-based lung nodule detection to ease the burden of lung nodule detection by radiologists. The purpose of this paper is to provide guidelines for collecting lung nodule data to facilitate research. We collected chest computed tomography scans reviewed by radiologists at three hospitals. In addition, several experiments were conducted using the large-scale open dataset, LUNA16. As a result of the experiment, it was possible to prove the value of using the collected data compared to using LUNA16. We also demonstrated the effectiveness of transfer learning from pre-trained learning weights in LUNA16. Finally, our study provides information on the amount of lung nodule data that must be collected to stabilize lung nodule detection performance.

**Abstract:**

Early detection of lung nodules is essential for preventing lung cancer. However, the number of radiologists who can diagnose lung nodules is limited, and considerable effort and time are required. To address this problem, researchers are investigating the automation of deep-learning-based lung nodule detection. However, deep learning requires large amounts of data, which can be difficult to collect. Therefore, data collection should be optimized to facilitate experiments at the beginning of lung nodule detection studies. We collected chest computed tomography scans from 515 patients with lung nodules from three hospitals and high-quality lung nodule annotations reviewed by radiologists. We conducted several experiments using the collected datasets and publicly available data from LUNA16. The object detection model, YOLOX was used in the lung nodule detection experiment. Similar or better performance was obtained when training the model with the collected data rather than LUNA16 with large amounts of data. We also show that weight transfer learning from pre-trained open data is very useful when it is difficult to collect large amounts of data. Good performance can otherwise be expected when reaching more than 100 patients. This study offers valuable insights for guiding data collection in lung nodules studies in the future.

## 1. Introduction

According to the World Health Organization, lung cancer is a life-threatening disease worldwide [1]. In 2018, lung cancer was the largest new cancer diagnosis (11.6%) and the largest cause of cancer-related deaths (18.4%) globally [2,3]. One of the main causes of high mortality in lung cancer is the absence of early overt clinical symptoms. The extent of lung cancer at the time of diagnosis is closely related to prognosis. Cancer Research UK reported that the survival rates for stages I and IV were 87% and 19% in patients with lung cancer, respectively [4]. Therefore, early detection and diagnosis are important for the curative treatment of lung cancer.

The National Lung Screening Trial showed that low-dose computed tomography (LDCT) reduces mortality by 20% compared with chest X-rays [5]. Posterior studies have shown that the early detection of lung nodules can increase the five-year survival rate of lung cancer patients. The five-year survival rate of lung cancer, which was estimated at 18% of all cases diagnosed between 2004–2010, is the lowest among several types of cancer [4]. However, manual detection of lung nodules in many CT scans is a very difficult, time-consuming, and tedious process, even for experienced radiologists [6,7,8]. Although chest CT is the latest and most widely used imaging tool to capture lung images [5], the manual detection of lung nodules by radiologists on CT images can easily lead to misdiagnosis [8]. An object detection system for lung nodules is necessary and several benchmark datasets have been used [9,10,11,12,13].

Consequently, artificial intelligence (AI)-assisted lung nodule detection systems may be used to provide a second opinion for radiologists and make final decisions faster and more accurately. Deep-learning models are highly dependent on datasets; thus, they can effectively achieve the performance of advanced learning algorithms when high-quality datasets are used for training [14]. Public databases such as LIDC/IDRI [9], LUNA16 [10], NSCLC [11], ELCAP [12], and ANODE09 [13] have been used for lung nodule diagnosis research. Nevertheless, these databases cannot be easily used to develop nodule detection systems because their format is not standardized. Therefore, it is important to create high-quality, real-world lung nodule datasets in a standardized format for training.

Recently, deep learning has been actively and successfully developed in various fields, such as image analysis, natural language processing, and speech recognition [15,16,17,18]. A convolutional neural network (CNN), a type of neural network that uses convolution operators, has been widely applied with excellent performance in image analysis. This success has contributed to the adoption of CNN in medical image analysis [19,20]. CNN performs well in deep learning technologies such as classification, object detection, and segmentation. In this study, we used CNN-based object detection to detect lung nodules by analyzing CT scan images. Object detection is a task that allows the drawing of a bounding box on a lung nodule to determine its exact location. CNN-based object detection tasks are making a lot of progress. Faster R-CNN [21], RetinaNet [22], EfficientDet [23], and YOLOX [24], etc., which were state-of-art models at the time, show the development of object detection. In this study, we experimented with lung nodule detection based on the YOLOX model. The specific details of the model used are described in Section 2.4.

It is necessary to train with a large amount of data to accurately detect lung nodules. However, it is difficult for most researchers to collect large amounts of medical data. To mitigate the difficulty of the lack of medical data, we conducted a study focusing on how much data should be collected at the beginning of a lung nodule study and what to do if it is difficult to collect a large amount of data. This study also considers how to leverage publicly available datasets when difficult to collect large amounts of data. Open datasets related to lung nodules [9,10,11,12,13] contain large amounts of data. However, open and privately collected data differ in many ways, for example in the information they offer on patient race and the way they annotate nodules. Therefore, open data are used; however, training that is suitable for private data is also required. Transfer learning is used to pre-train models on open data and transfers that knowledge to private data [25]. In this study, we demonstrated the usability of open datasets and the importance of building private datasets through various experiments using both open and private data. 

The main contributions of this study are: (1) The construction of a high-quality chest CT scan dataset reviewed by radiologists from three hospitals for lung nodule detection; (2) The demonstration of the importance of collecting private data by comparing open data with collected private data; (3) The demonstration that transfer learning using open data is highly effective when a large amount of data cannot be collected; (4) Through empirical experiments, the provision of insights into the necessary amount of data to collect. 

## 2. Materials and Methods

### 2.1. Datasets

#### 2.1.1. LUNA16

LUNA16 is a publicly available dataset for lung nodule detection and a subset of the LIDC/IDRI dataset. The LIDC/IDRI dataset contains 1018 CT scans and annotations confirmed by four experienced radiologists. The radiologists reviewed each CT scan and marked the lesions in three categories: nodules ≥ 3 mm, nodules < 3 mm, and non-nodules. The LUNA16 dataset used in this study only considers nodules ≥ 3 mm in diameter. The LUNA16 dataset also excludes CT scans with a slice thickness greater than 3 mm, inconsistent slice spacing, and missing slices from the LIDC/IDRI dataset. The LUNA16 dataset contains 888 chest CT scans and annotations consisted of the centroid coordinates and diameters of lung nodules.

#### 2.1.2. Private Dataset

We retrospectively reviewed the medical records and contrast-enhanced chest CT scans of patients with solitary pulmonary nodules (SPN) that were followed up at Gangneung Asan Hospital (GNAH), Kangwon National University Hospital (KNUH), and Hallym Sacred Heart Hospital (HSHH) between March 2016 and February 2021. The private dataset was reviewed by radiologists in three hospitals, and chest CT scan data were collected from 515 patients with lung nodules (GNAH, 287; KNUH, 181; HSHH, 47). We used the Digital Imaging and Communications in Medicine (DICOM) file format.

There were four inclusion criteria: (1) the patient was aged between 18 and 90 years; (2) the nodule diameter in the chest CT axial image was more than 7 mm and less than 30 mm [26]; (3) the pathological diagnosis was performed using percutaneous needle aspiration (PCNA) biopsy, surgical operation, bronchoscopic biopsy, or endobronchial ultrasonography-guided transbronchial needle aspiration (EBUS-TBNA); and (4) in the absence of a pathological diagnosis, follow-up was performed with contrast-enhanced chest CT scan for more than two years. Table A1, Table A2 and Table A3 in Appendix A showed the clinical data of the private dataset that explained inclusion criteria. In the NSCLC category in Table A2 and Table A3, the sub-NSCLC category is NSCLC with pathological findings that do not fall into any category of adenocarcinoma, squamous cell carcinoma, or large cell endocrine carcinoma.

In this study, we defined an SPN as benign when (1) the pathological diagnosis was confirmed; and (2) there was no increase in size, or there was disappearance of SPN when followed up by a chest CT scan for more than two years [27]. An SPN was defined as malignant only when the pathological diagnosis was confirmed.

The three datasets were formed as follows, using the chest CT protocols outlined in Table 1:GNAH dataset: 287 nodules (102 benign SPNs, and 185 malignant SPNs);KNUH dataset: 181 nodules (36 benign SPNs, and 145 malignant SPNs);HSHH dataset: 47 nodules (17 benign SPNs, and 30 malignant SPNs).

#### 2.1.3. Annotation

Using LUNA16 and private data, we created bounding boxes of lung nodules, which were used as the ground truth of the object detection task. To determine the coordinates of a bounding box in LUNA16, we used an annotation consisting of the centroid coordinates and diameters of the 3D nodules. The bounding boxes and annotations of private data were created with the help of the experts at each hospital. To this end, we first obtained the nodule centroid coordinates for each slice as identified by an expert. Based on the nodule centroid coordinates, we constructed a lung nodule bounding box using a computer vision annotation tool (CVAT). CVAT is an open-source platform that can build and share private servers. We then shared the lung nodule bounding boxes with experts to review and modify them. Using the bounding boxes in LUNA16 and our private data, we prepared annotations in pascal visual object classes challenge (VOC) format, which is a representative object detection data format [28]. The annotation of the VOC format consists of the filename, size, object name, and coordinates of the bounding box. Figure 1 shows a set of bounding boxes in LUNA16 and the private data. Figure 1c,d corresponding to the LUNA16 dataset show how the bounding box can at times represent the nodule position while having a size larger than that of the nodule. These cases negatively affect the detection of sophisticated boxes, an issue that we will further describe in the Results.

#### 2.1.4. Dataset Configuration

The LUNA16 dataset consisted of 10 subsets. We used eight subsets (712 patients) for the training dataset and two subsets (176 patients) for the validation dataset. The private dataset included a total of 515 CT scans consisting of 181 data from KNUH, 287 data from GNAH, and 47 data from HSHH. To learn from historical data and test relatively recent data, we sorted each hospital’s data by chronological order based on the date of the CT scan. The GNAH and KNUH data were accordingly divided into 60% of the data for training, 20% of the data for validation (a dataset used to tune the hyper parameters of a detector), and 20% of the data for testing (a dataset used only to assess the performance of a fully specified detector) (Figure 2). The HSHH dataset was used as an external validation set, which was used only for testing, with no involvement in training and validation.

### 2.2. Preprocessing

Unlike normal image data with values of 0–255, CT scan data consist of 10–16 bits of Hounsfield units (HU). To use CT data for deep learning, we thus need to set their values within the 0–1 range. Certain tissues and organs, such as the lung, brain, bone, and skin, generally have a constant range of HU values and adjusting the width center (W_C_) and the window width (W_W_) can emphasize the areas of interest. W_C_ is the HU value of the specific area to be emphasized and W_W_ is the HU range to be observed. In this study, we used window-setting values suitable for lung nodule detection (W_C_ = −700; W_W_ = 1500) [29,30,31,32]. The range of the CT scan data can be defined as:(1)$$X_{\mathrm{CT}}\in \left[W_{c}-\frac{W_{w}}{2},\;{\;W}_{C}+\frac{W_{w}}{2}\;\right]$$

We clipped the CT scan data into (−1450, 50) using the minimum and maximum values obtained through Equation (1). Then, the range of the CT scan data was linearly normalized to (0, 1), as follows:(2)$${\tilde{X}}_{\mathrm{CT}}=\frac{X_{\mathrm{CT}}-min\left(X_{\mathrm{CT}}\right)}{max\left(X_{\mathrm{CT}}\right)-min\left(X_{\mathrm{CT}}\right)}$$

### 2.3. Data Augmentation

Data augmentation generates artificial data to increase the amount of data [33]. This technology alleviates the overfitting of deep-learning models. Data augmentation is essential in the medical field where large amounts of data are difficult to obtain. We applied the following augmentation techniques to the training model in the following order: (1) flip; (2) mosaic [34]; (3) random affine transformation; and (4) MixUp [35]. After doing image flipping with a probability of 0.5, we applied a mosaic, a strong data augmentation technique that combines up to four images into one. In addition to learning from four images with one input, because the number of small objects increases in the process of combining the images, the detection performance of small objects can be expected to improve. After mosaic application, random affine transformations such as rotation, translation, and scaling were applied. Finally, MixUp, in which two images are mixed at a certain ratio, was used to produce a single image. MixUp helps generalize deep learning models. Figure 3 shows an example of applying data augmentation in LUNA16.

### 2.4. Object Detection and Deep Learning Model

In this study, an object detection algorithm, which is a deep learning technique, was used to detect lung nodules in lung CT scans. Object detection is the task of classifying and localizing all the objects present in an image. Localization is a regression operation that finds a bounding box, whereas classification is an operation that classifies objects in a bounding box. 

We used YOLOX to detect lung nodules [24]. The architecture of YOLOX is depicted in Figure 4. YOLOX is an object detection model with excellent accuracy and speed that simultaneously applies various detection techniques. This model is a one-stage detector and consists of a backbone-neck decoupled head. YOLOX extracts a feature map from the input image using a darknet-53 backbone. The neck connects the backbone and the head and consists of spatial pyramid pooling (SPP) and feature pyramid network (FPN) [36,37]. SPP improves performance by playing a role in maintaining spatial structure information. FPN receives a feature map extracted from the backbone and obtains a multiscale feature map. FPN adds semantic information preserved by a high-level layer to a low-level layer to make it robust against scale variations. The feature map obtained from FPN enters the decoupled head and finds the class and box of the object to derive the detection results. The decoupled head improves the performance by separating the classification and regression heads to solve the problem caused by the conflict between classification and regression work [38]. In addition, YOLOX uses various technologies, such as anchor-free, SimOTA, and multi-positive, to improve performance.

### 2.5. Evaluation Metrics

In general, we computed true positives (TP), false positives (FP), false negatives (FN), and true negatives (TN) by comparing the predicted and the ground truth to measure the predictive performance of the trained model. In this study, TP refers to instances where the deep learning model correctly predicted lung nodules. FP refers to cases where the deep learning model incorrectly predicted non-lung nodules as lung nodules. FN refers to instances where the deep learning model incorrectly predicted lung nodules as non-lung nodules. Finally, TN refers to cases where the deep learning model correctly predicted that non-lung nodules were not lung nodules. In the object detection task, the intersection over union (IOU) of the predicted bounding box and the actual bounding box were calculated. If the IOU was above the threshold value, the bounding box was judged to be true, and if IOU was under the threshold value, the bounding box was judged to be false. Sensitivity and precision can be calculated using this information. The sensitivity is the ratio of the lung nodules that were correctly predicted to the actual lung nodules. The precision is the ratio of the correctly predicted lung nodules to all predicted lung nodules. Finally, based on this information, lung nodule detection performance was evaluated using the average precision (AP) and competition performance metric (CPM) [39] utilizing free-response receiver operating characteristic (FROC). AP and CPM were used as quantitative evaluation metrics. AP is an evaluation metric that considers both precision and sensitivity and is widely used to evaluate the performance of object detection algorithms. Lung nodules are very small objects, and the IOU values are very low, even if the size or position of the box is slightly different from the ground truth. For this reason, even an actual true box can be judged false if the IOU threshold is high. Therefore, the performance measurements were carried out using AP10 (AP at IOU = 0.1) and AP50 (AP at IOU = 0.5). The AP10 metric has the advantage of having a high sensitivity because it is recognized as true, even if the overlap between the prediction and ground truth is low. The AP50 metric may have a lower sensitivity performance because the IOU of the prediction and ground truth must be 0.5 or more to be true, but more sophisticated lung nodule bounding box performance can be measured. FROC analysis is utilized for nodule detection evaluation in the LUNA16 challenge. In the FROC curve, sensitivity is plotted as a function of the average number of false positives per scan (FPs/scan). The CPM score is the average sensitivity at seven predefined false-positive rates (1/8, 1/4, 1/2, 1, 2, 4, and 8FPs per scan).

### 2.6. Dataset Configuration and Experimental Strategy 

The dataset configurations used in the experiments are presented in Table 2. We prepared five configurations using both open and private datasets for training and validation, whereas for testing we used only one configuration using private datasets, GNAH (20%), KNUH (20%), and HSHH (100%). In configuration (A), we used only the open dataset LUNA16 and eight subsets for training and two subsets for validation. In configuration (B), we used LUNA16 for training and two private datasets (GNAH and KNUH) for validation. In the other configurations (C, D, and E), we used only private datasets for both training and validation.

To analyze the effects of open and private datasets, we then conducted three experiments depending on the settings of the LUNA16 and private datasets (Figure 5). Experiment 1 was conducted to compare the usefulness of the open dataset and private datasets when used for training. Experiment 2 was conducted to evaluate the effect of pre-training using LUNA16 before training with private data. In Experiment 3, performance improvement was measured by gradually increasing private data to analyze the amount of private data required. All experiments were conducted on an NVIDIA GeForce GTX 1080Ti GPU with 11GB of graphics memory. We trained nodule detection models using the Pytorch framework. For the detection model, we used YOLOX-S. In the training, we used a stochastic gradient descent (SGD) optimizer with a momentum of 0.9, a weight decay of 0.0005, and a learning rate of lr× batch size/64. The initial lr was 0.01 and the batch size was set to 16. The learning rate scheduler adopts a cosine LR schedule. The input size was 640 × 640 pixels. The model was then trained for 200 epochs. All experiments were conducted using a fixed seed = 1.

## 3. Results

### 3.1. Experiment 1: Comparing the Performance of Collected Private Data and Open Data

This experiment compared the performance of a model trained on the LUNA16 dataset with that of a model trained on a private dataset. Table 3 summarizes the performance obtained in Experiment 1, demonstrating that learning from private data led to a competitive performance of the model compared to that obtained when learning from LUNA16. In addition, when the same hospital data were used for both training and testing, the performance was better than that obtained when using LUNA16 for training. Furthermore, the AP50 performance of the model trained on private data was significantly improved compared with that of the model trained on LUNA16. Moreover, the performance using the combined dataset of GNAH and KNUH was higher than that using the individual hospital datasets. This result suggests that collecting a large amount of private data is important for performance improvement.

Figure 6 shows an image superimposing the prediction results of the models trained for the ground truth and LUNA16, as well as the prediction results of the models trained for private data. As shown in Figure 6b,c, the prediction results of the LUNA16 training model tended to draw a box larger than the lung nodule, suggesting a difference in the performance of AP50. Figure 6d shows a case in which the private data training model predicted the lung nodules unforeseen in the LUNA16 training model.

### 3.2. Experiment 2: Is Pre-Training Using Open Data Effective?

We found that although training on LUNA16 only in Experiment 1 was not sophisticated, the model learned the characteristics of lung nodules well enough to predict their locations. Hence, we can expect pre-training on LUNA16 to have a positive effect on training with private data. To confirm this, we experimented with a transfer learning method using pre-training on LUNA16. We compared the cases with and without transfer learning that adopted weights pre-trained on LUNA16. Figure 7 shows the CPM results from Experiment 2. Most of the performances were higher when applying transfer learning. However, when the combined data from GNAH and KNUH were used for training, the performance was higher when transfer learning was not applied. The resulting graphs for Experiment 2 in its entirety are shown in Figure A1 in Appendix B.

### 3.3. Experiment 3: How Much Data Do I Need to Collect?

Experiment 3 evaluated performance while gradually increasing the amount of private data. The data were sorted based on the date of the CT scan, and old data were extracted first. The combined data of GNAH and KNUH were gradually increased with 10 patients from GNAH and 10 patients from KNUH and the performance was evaluated. We also trained and evaluated the EfficientDet [23] model as well as the YOLOX model for the generalization of the conclusion (Figure 8). When using a small amount of data without transfer learning, the performance was very poor. However, when using approximately 100 patient data points, the performance became similar to the case using transfer learning. When transferring knowledge from open data, high performance could be achieved, even with a small amount of data. Additional graphs for Experiment 3 are shown in Figure A2 and Figure A3 of Appendix B.

## 4. Discussion

Public datasets are essential for the better application of deep learning to medical problems in clinical settings [40]. Identifying issues and biases in commonly used benchmark datasets requires the use of private datasets [41,42]. In this study, several experiments were conducted using publicly available LUNA16 data and private data collected from three hospitals.

In Experiment 1, we were able to compare the results when training the model on LUNA16 and on private data. Compared to training the model on LUNA16 with large numbers of CT scans, private data collected in each hospital led to a similar performance in lung nodule detection. Using training and testing data from the same hospital led to higher performance than when training on LUNA16. These results indicate that using training data from the same hospital can be expected to lead to good performance in the detection of lung nodules in private hospitals. In Experiment 1, it was confirmed that even if the model was trained on LUNA16 alone, it was possible to identify and detect lung nodule features in private test data. However, the annotation of LUNA16 consists only of the central coordinates and diameters of the 3D nodules, not the bounding box coordinates for each slice, making it impossible to create a sophisticated box for each slice. Private data included slice-by-slice boxes reviewed by radiologists as annotations, allowing sophisticated boxes to be detected. The AP50 results showed that learning on private data can yield more sophisticated results than learning on LUNA16 only. Finally, training could be done by combining the training datasets of KNUH and GNAH. These results show that a good performance can be expected for lung nodule detection if a dataset is built by collecting a large amount of data.

In Experiment 2, the performance of transfer learning based on the weight pre-trained on LUNA16 and the performance without transfer learning were compared. When training on individual hospital data, most of the cases of transfer learning were better than those that did not. However, the performance was found to be better without transfer learning when the model was trained using KNUH in the HSHH-Test. This phenomenon appears to be due to differences in the CT slice thickness between the datasets [43]. The KNUH and HSHH datasets had the same slice thickness (3 mm). For this reason, even though only KNUH data were used for training, the performance may be higher than that of transfer learning when HSHH data are tested. In this case, it seems that applying transfer learning in the HSHH-test is a factor reducing the performance. These results show that if the CT protocol is similar and the slice gap is the same, the performance is good when testing the CT data under the same conditions. Moreover, AP10 and CPM performance were higher when applying transfer learning to graphs trained with GNAH in the GNAH-TEST, but the AP50 performance was lower. In this case, the location of the lung nodule was better identified when applying transfer learning, but the sophistication of the nodule box was reduced. In the future, it will be necessary to collect more private data and analyze these cases in depth. When using the training dataset combining KNUH and GNAH, the performance improved without transfer learning. These results demonstrate that high performance can be achieved without applying transfer learning if sufficient data can be collected.

Experiment 3 tested the amount of data that needed to be collected to detect the lung nodules. Performance was measured as the amount of private data gradually increased, starting with the data from 20 patients. When no transfer learning was performed, the performance was very poor when training was performed with a small amount of data. However, when close to 100 patient data points were collected, the performance was similar to or increased more strongly than when the model was trained on LUNA16. When LUNA16 transfer learning was performed under the same conditions, the performance was high, even with a small amount of private data training. Transfer learning from LUNA16 is very effective in situations where it is not possible to collect a large amount of data.

To summarize the results, to evaluate AI algorithms for detecting lung nodules of chest CT scans in private hospitals, it is better to collect and build a private dataset that can produce more sophisticated results than using only open data. However, if it is difficult to collect a large amount of private data, it is effective to utilize pre-trained weights with open data. If a large amount of private data can be collected, good performance can be expected without transfer learning. These results provide guidelines for building the private datasets necessary for the model to learn lung nodules.

To the best of our knowledge, this paper is the first to determine the amount of private data required to perform object detection for an SPN on an enhanced chest CT scan. Because it is a multicenter study of three hospitals, it can better represent real-world datasets and their heterogeneity than public datasets. The size of private datasets was sufficient to perform an object detection model without using public data. Most recently, Xu et al. [44] reported a YOLOv3 network for detecting lung nodules on chest CT using 314 private data points. However, our private dataset of 515 patients was larger than Xu’s private dataset.

Although this study obtained meaningful results through lung nodule detection experiments using data from various hospitals, it had several limitations. First, we did not study a technology to improve lung nodule detection performance. For further study, we will focus on enhancing detection performance itself by adapting state-of-the-art techniques such as generative adversarial networks-based augmentation. Second, the classification of benign and malignant nodules was not considered because only the task of detecting the location of the lung nodules was performed.

This study shows that when detecting lung nodules from private data, training the model on private data collected from the same hospital can yield more sophisticated results than training the model on massive open data. Moreover, the performance based on the amount of private data was confirmed, and it was found that transfer learning had a significant influence when the amount of data was small. However, the effect of transfer learning was reduced if a large amount of data was collected and used for training. Therefore, if a large amount of data is collected, good performance can be expected without having to transfer the weights pre-trained on the open data.

The final goal of our study is to present guidelines on how many private data should be collected when conducting a study to detect lung nodules with chest CT images. Most deep learning researchers who want to analyze medical data use public data because it is difficult to obtain private data. However, as the reviewer has already mentioned, public data does not reflect the characteristics of data based on the real world, such as when a CT machine is changed, a new protocol is released, or the definition of a pulmonary nodule is changed. This is because even if a deep learning model is developed with public data, it cannot be applied to private data for the same reason as above, and training suitable for private data may need to be renewed again. Additionally, the issues of whether artificial intelligence can replace radiologists and who should be held responsible for making a wrong decision are still controversial, and I think that AI researchers’ consensus is needed to solve these problems. At least for now, it should be considered as one of the diagnostic aids that reduce the fatigue of medical staff.

## 5. Conclusions

The experimental results of this study will help researchers study lung nodule detection for the collection of private data and obtaining results. In future research, we plan to build larger datasets with continuous data supply and demand. Unlike ordinary images, medical images must be labeled by specialists. The more data there are, the more labeling is needed, but the number of specialists is limited. Therefore, it is necessary to increase the time spent by specialists and the efficiency of the labor force. We plan to study active learning algorithms to proceed efficiently with data labeling. Active learning allows the selection of only the data with high uncertainty and requires labeling from a specialist. Consequently, data can be smoothly built by reducing the amount of data required for labeling. In addition, we plan to study AI technologies that are suitable for the characteristics of chest CT scans and to conduct research that can contribute to engineering. First, it will be valuable to experiment by applying various window settings for chest CT scanning. Depending on the window setting, the range of HUs can be adjusted to obtain various chest CT images. It is also possible to consider combining images as inputs into deep learning models using multiple sets of lung CT images or combining the results of each image. Second, we are starting to use object detection models using slices, which include box annotation; therefore, there is a lot of data that remains unused. We also plan to conduct a study using CT slices that do not include lung nodules to devise anomaly detection technology. Finally, because there are many slices containing lung tissues among the unused data, research on lung detection or lung segmentation can also be a good approach.

## Figures and Tables

**Figure 1 cancers-14-03174-f001:**
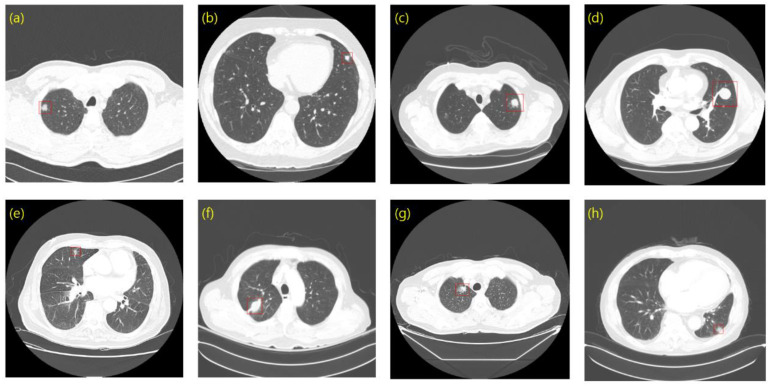
Ground truth sample. (**a**–**d**) LUNA16; (**e**–**h**) Private data.

**Figure 2 cancers-14-03174-f002:**
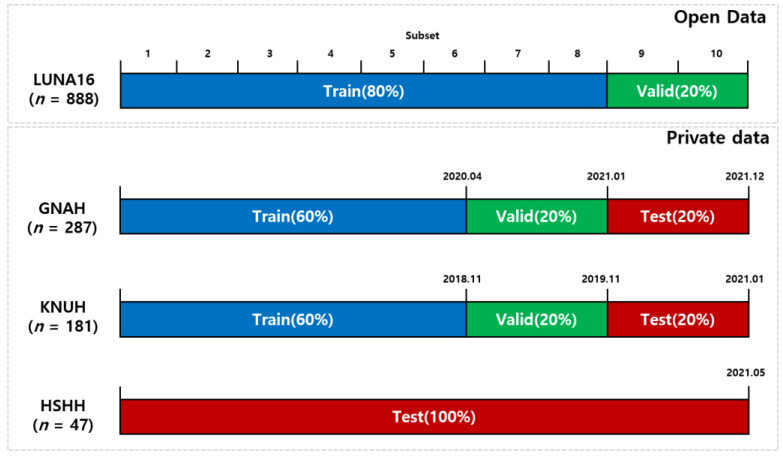
Illustration of dataset configuration.

**Figure 3 cancers-14-03174-f003:**
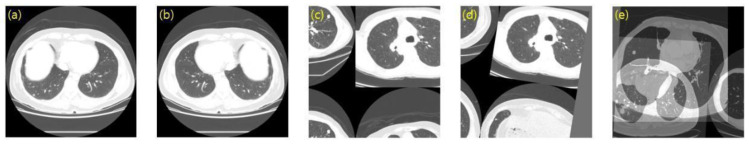
Example of data augmentation applied to lung nodule CT images. (**a**) Original; (**b**) Flip; (**c**) Mosaic; (**d**) Random Affine; (**e**) MixUp.

**Figure 4 cancers-14-03174-f004:**
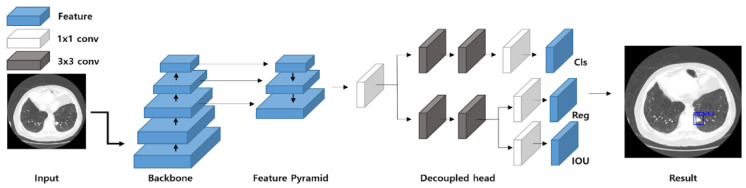
Detailed architecture of YOLOX. Cls, Class; Reg, Regression; IOU, intersection over union; conv, convolution.

**Figure 5 cancers-14-03174-f005:**
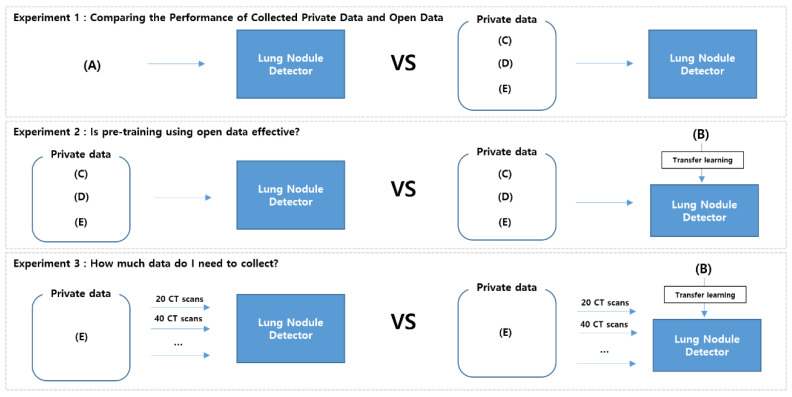
Experimental settings.

**Figure 6 cancers-14-03174-f006:**
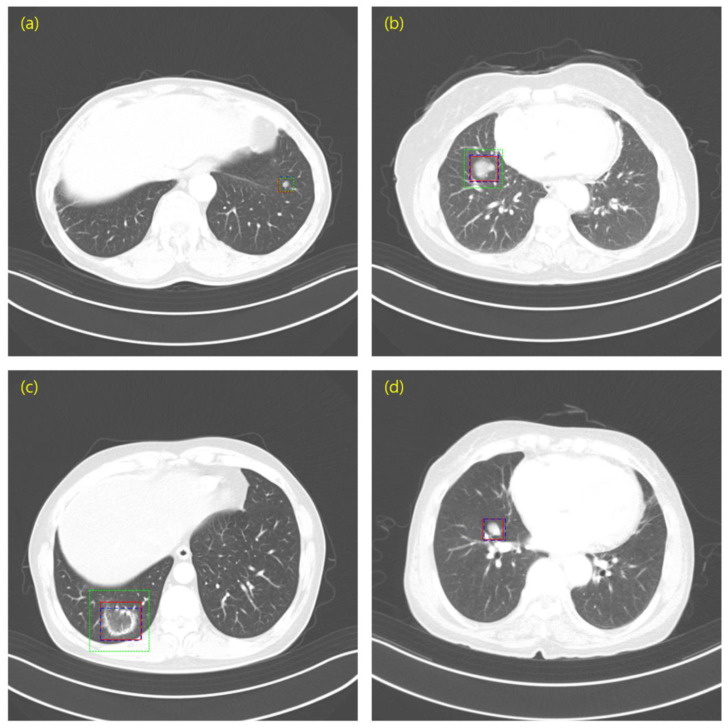
Image sample overlapping the prediction results of the LUNA16 and private data trained models and the ground truth. (**a**) Case where the predictions of the model trained on LUNA16 and the model trained on private data is very similar to the ground truth; (**b**,**c**) Case where the predicted box of the model trained on LUNA16 is significantly larger than the size of the ground truth; (**d**) Case in which the LUNA16 trained model did not predict lung nodules. Red (Solid line) = Ground Truth; Blue (Dashed line) = Private data; Green (Dotted line) = LUNA16.

**Figure 7 cancers-14-03174-f007:**
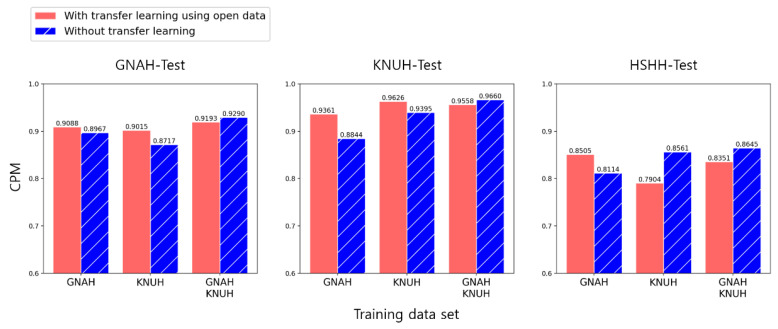
Comparison of the performance of the models with transfer learning from open data and without transfer learning.

**Figure 8 cancers-14-03174-f008:**
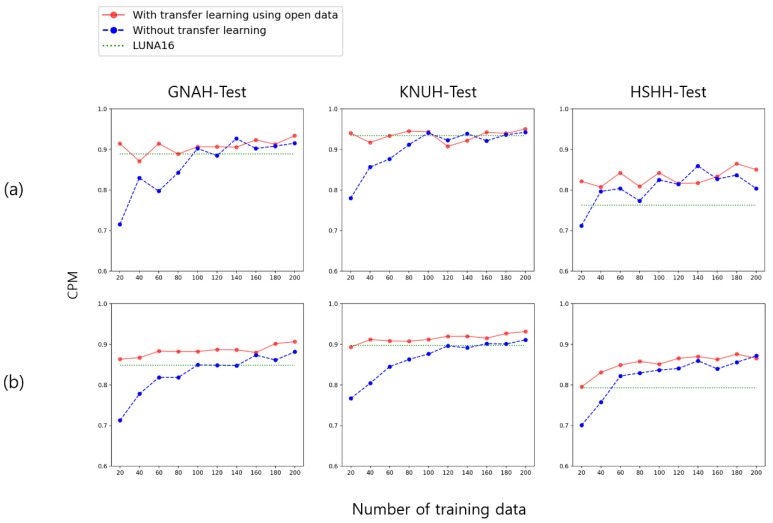
CPM results for models trained with incremental growth of GNAH and KNUH combined data. (**a**) YOLOX; (**b**) EfficientDet.

**Table 1 cancers-14-03174-t001:** Chest CT protocols at Gangneung Asan Hospital (GNAH), Kangwon National University Hospital (KNUH), and Hallym Sacred Heart Hospital (HSHH).

CT Protocols	GNAH Dataset	KNUH Dataset	HSHH Dataset
Name	SIEMENS/SOMATOM Definition Edge 2 (128 ch)	SIEMENS/SOMATOM Definition & Definition Flash	SIEMENS/SOMATOM Flash (128 ch)
kVp/mAs	120/35	120/110	120/35
kernel	B41f medium	B41f	B40f medium
slice/gap (mm)	5	3	3

**Table 2 cancers-14-03174-t002:** Dataset configurations used in the experiments.

Dataset	Training	Validation	Test
(A)	LUNA16 (subsets 1–8)	LUNA16 (subsets 9, 10)	GNAH (20%) KNUH (20%) HSHH (100%)
(B)	LUNA16 pre-train (100%)	GNAH (20%) KNUH (20%)	
(C)	GNAH (60%)	GNAH (20%)	
(D)	KNUH (60%)	KNUH (20%)	
(E)	GNAH (60%), KNUH (60%)	GNAH (20%) KNUH (20%)	

**Table 3 cancers-14-03174-t003:** Performance Comparison of Collected Private Data and Open Data.

Training Dataset	Test Dataset	AP10	AP50	CPM
LUNA16	GNAH	0.8590 (+0.0)	0.5482 (+0.0)	0.8886 (+0.0)
	KNUH	0.9151 (+0.0)	0.4499 (+0.0)	0.9340 (+0.0)
	HSHH	0.7262 (+0.0)	0.1111 (+0.0)	0.7628 (+0.0)
GNAH	GNAH	0.8674 (+0.0084)	0.8160 (+0.2678)	0.8967 (+0.0081)
	KNUH	0.8548 (−0.0603)	0.8228 (+0.3729)	0.8844 (−0.0496)
	HSHH	0.7878 (+0.0616)	0.6270 (+0.5159)	0.8114 (+0.0486)
KNUH	GNAH	0.8455 (−0.0135)	0.7996 (+0.2514)	0.8717 (−0.0169)
	KNUH	0.9167 (+0.0016)	0.8817 (+0.4318)	0.9395 (+0.0055)
	HSHH	0.8307 (+0.1045)	0.7729 (+0.6618)	0.8561 (+0.0933)
GNAH ∪ KNUH	GNAH	0.8960 (+0.0370)	0.8450 (+0.2968)	0.9290 (+0.0404)
	KNUH	0.9525 (+0.0374)	0.9341 (+0.4842)	0.9660 (+0.0320)
	HSHH	0.8391 (+0.1129)	0.7935 (+0.6824)	0.8645 (+0.1017)

AP, average precision; CPM, competition performance metric.

## Data Availability

The data presented in this study are available upon request from the corresponding author.

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
