# Peer review of "How Many Private Data Are Needed for Deep Learning in Lung Nodule Detection on CT Scans? A Retrospective Multicenter Study"

_cancers, 2022, doi:10.3390/cancers14133174_

Round 1

Reviewer 1 Report

In the manuscript a study about the amount of images necessary to train a CNN for lung cancer detection is conducted. The experiments include a public dataset, LUNA16 and three private dataset. Extensive experiments were conducted varying the training dataset, performing transfer learning and increasing the amount of private data. Moreover, an interesting strategy of augmentation of the data was made. The results are promising and they confirm the hypotesis that the use of private dataset from variuos institutions increase the performance of a Deep Learning approach and this also help to generalize the model. 

The manuscript is well written and in fluent english. I suggest to modify only Discussion section. In details:

- The first paragraph from line 346 to line 355 does not appear to have been written in the correct place. I suggest to move it in the introduction and leave in the Discussion only the part related to discussions on the results;

- In this section, the word "study" is repeated too many times;

- Finally, I suggest to move in the "Conclusions" the part related to future works.

Author Response

We would like to thank you and the reviewers of the Cancers for taking the time and effort to review our manuscript. Many of the valuable and constructive points that the reviewers raised, were appreciated by the authors. After considering the reviewers’ comments, we revised the manuscript and have indicated the corrections and changes made with yellow highlighting in the manuscript.

- The first paragraph from line 346 to line 355 does not appear to have been written in the correct place. I suggest moving it in the introduction and leave in the Discussion only the part related to discussions on the results.

Reply 1: We greatly appreciate the Reviewer’s comments and agree completely. As recommended, we removed the first paragraph in the Discussion (from line 346 to line 355), and moved it in the Introduction as followed:

“The National Lung Screening Trial showed that low-dose computed tomography (LDCT) reduces mortality by 20% compared with chest X-rays. Posterior studies have shown that the early detection of lung nodules can increase the 5-year survival rate of lung cancer patients. The 5-year survival rate of lung cancer, which was estimated at 18% of all cases diagnosed between 2004–2010, is the lowest among several types of cancer. However, manual detection of lung nodules in many CT scans is a very difficult, time-consuming, and tedious process, even for experienced radiologists Although chest CT is the latest and most widely used imaging tool to capture lung images, the manual detection of lung nodules by radiologists on CT images can easily lead to misdiagnosis. An object detection system for lung nodules is necessary and several benchmark datasets have been used.”

- In this section, the word "study" is repeated too many times.

Reply 2: We greatly appreciate the Reviewer’s comments. We have modified the text as followed:

“To summarize the results, to evaluate AI algorithms for detecting lung nodules of chest CT scans in private hospitals, it is better to collect and build a private dataset that can produce more sophisticated results than using only open data.”

“These results provide guidelines for building the private datasets necessary for the model to learn lung nodules.”

“To the best of our knowledge, this paper is the first to determine the amount of private data required to perform object detection for an SPN on an enhanced chest CT scan.”

“Because it is a multicenter study of three hospitals, it can better represent real-world datasets and their heterogeneity than public datasets.”

“Although this study obtained meaningful results through lung nodule detection experiments using data from various hospitals, it had several limitations. First, we did not study a technology to improve the lung nodule detection performance.”

- Finally, I suggest moving in the "Conclusions" the part related to future works.

Reply 3: We appreciate the Reviewer’s helpful suggestion. We agree with the Reviewer’s comment and have revised as followed in the Conclusions:

“The experimental results of this study will help researchers to study lung nodule detection for the collection of private data and obtaining of results. In future research, we plan to build larger datasets with continuous data supply and demand. Unlike ordinary images, medical images must be labelled by specialists. The more data, the more labeling is needed, but the number of specialists is limited. Therefore, it is necessary to increase the time spent by specialists and the efficiency of the labor force. We plan to study active learning algorithms to proceed efficiently with data labeling. Active learning allows the selection of only the data with a high uncertainty and requires labeling from a specialist. Consequently, data can be smoothly built by reducing the amount of data required for labeling. In addition, we plan to study AI technologies that are suitable for the characteristics of chest CT scans and to conduct research that can contribute to engineering. First, it will be valuable to experiment by applying various window settings for chest CT scanning. Depending on the window setting, the range of HUs can be adjusted to obtain various chest CT images. It is also possible to consider combining images as inputs into deep learning models using multiple sets of lung CT images or combining the results of each image. Second, we are starting to use object detection models using slices, which include box annotation; therefore, there is a lot of data that remains unused. We also plan to conduct a study using CT slices that do not include lung nodules to devise anomaly detection technology. Finally, because there are many slices containing lung tissues among the unused data, research on lung detection or lung segmentation can also be a good approach.”

Reviewer 2 Report

The authors present an interesting article about automatic detection of lung nodules by trained neuronal networks. They try to answer the central question in this field: How can one get enough and sufficient good data for the training of the network. As a result, they find some practical recommendations.

But there are two issues that should discussed in more details. First, the final goal is not clear. Shall the radiologist be substituted? Who is then responsible for a wrong decision? A very important aspect. Alternatively, shall the trained network serve as an additional helpful instrument for detecting lung nodules? What can be the advantages in this case? How large can be the relief? Another point that should be pronounced is that after relevant changes (new CT, better resolution of the images or new criterions for relevant nodules) a new training of the system could be necessary.

Further remarks:

·         * In the LUNA16 data, nodules are considered as equal or larger than 3mm. In the private data, however as larger than 7mm. Could this be the reason that the training on LUNA16 data is worse than the training on private data in experiment 1? What is the influence of these different “nodule definitions”?

·         * The authors should give a short explanation of the difference between validation and test.

·         * In figure 5, maybe A and B have to be exchanged.

·         * In appendix B, it would be better to change the names of figures A1 and A2 to figures B1 and B2.

·        *  Page 2, line 10: databases instead of databased?

Author Response

Dear Reviewer 2.

We would like to thank you and the reviewers of the Cancers for taking the time and effort to review our manuscript. Many of the valuable and constructive points that the reviewers raised, were appreciated by the authors. After considering the reviewers’ comments, we revised the manuscript and have indicated the corrections and changes made with yellow highlighting in the manuscript.

Comments to the Author

The authors present an interesting article about the automatic detection of lung nodules by trained neuronal networks. They try to answer the central question in this field: How can one get enough and sufficient good data for the training of the network. As a result, they find some practical recommendations.

But there are two issues that should be discussed in more detail.

  • First, the final goal is not clear. Shall the radiologist be substituted? Who is then responsible for a wrong decision? A very important aspect. Alternatively, shall the trained network serve as an additional helpful instrument for detecting lung nodules? What can be the advantages in this case? How large can be the relief? Another point that should be pronounced is that after relevant changes (new CT, better resolution of the images, or new criteria for relevant nodules) a new training of the system could be necessary.

Reply 1: We greatly appreciate the Reviewer’s comments. We agree completely and have revised as followed in the Discussion:

“The final goal of our study is to present guidelines on how much private data should be collected when conducting a study to detect lung nodules with chest CT images. Most of deep learning researchers who want to analyze medical data use public data because it is difficult to obtain private data. However, as the Reviewer has already mentioned, public data does not reflect the characteristics of data based on the real world, such as when a CT machine is changed, a new protocol is released, or the definition of a pulmonary nodule is changed. This is because even if a deep learning model is developed with public data, it cannot be applied to private data for the same reason as above, and training suitable for private data may need to be renewed again. And the issues of whether artificial intelligence can replace radiologists and who should be held responsible for making a wrong decision are still controversial, and I think that AI researchers' consensus is needed to solve these problems. At least for now, it should be considered as one of the diagnostic aids that reduce the fatigue of medical staff.”

Further remarks:

  • In the LUNA16 data, nodules are considered as equal or larger than 3mm. In the private data, however as larger than 7mm. Could this be the reason that the training on LUNA16 data is worse than the training on private data in experiment 1? What is the influence of these different “nodule definitions”?

Reply 2: We greatly appreciate the Reviewer’s comments. In Table 3, when testing on GNAH's private data, AP10 and CPM's results show that the case of training with KNUH's private data is worse than the case of training with LUNA16's public data. Therefore, we may think that it is difficult to explain this only with the difference between the “nodule definition” of private data and public data.

  • The authors should give a short explanation of the difference between validation and test.

Reply 3: We appreciate the Reviewer’s helpful suggestion. We agree with the Reviewer’s comment and have revised as followed in the Materials and Methods:

“The GNAH and KNUH data were accordingly divided into 60% of the data for training, 20% of the data for validation (a dataset used to tune the hyperparameters of a detector), and 20% of the data for testing (a dataset used only to assess the performance of a fully specified detector) (Figure 2).

  • * In figure 5, maybe A and B have to be exchanged.

Reply 4: We appreciate for the precise review. We agree review’s comment and have revised the figure 5 (A and B have changed).

  • In appendix B, it would be better to change the names of figures A1 and A2 to figures B1 and B2.

Reply 5: We greatly appreciate the Reviewer’s comments. We have revised the names of figures A1 and A2 to figures B1 and B2.

  • Page 2, line 10: databases instead of databased?

Reply 6: We apologize for this typo. We have corrected as followed in the Introduction:

“Nevertheless, these databases cannot be easily used to develop nodule detection systems because their format is not standardized.”

Sincerely,

The authors.

Reviewer 3 Report

Please refer to the attached PDF

I am glad to review the manuscript entitled How Much Private Data Is

Needed for Deep Learning in Lung Nodule Detection on CT scans? A

Retrospective Multicenter Study. This manuscript provides guidelines for

collecting lung nodule data to facilitate research. Although there are

some major problems which have to be addressed through a significant

revision, the overall quality of the research is satisfactory.

In this manuscript, the authors set up five experimental configurations

to solve three problems in the process of lung nodule data collection.

Firstly, the authors compare the performance of a model trained on the

LUNA16 dataset with that of a model trained on a private dataset. Then

they confirm that pre-training on open data has a positive effect by

comparing the results with and without transfer learning. Finally, the

authors demonstrate the model performance with the different amount of

private data.

This manuscript did a very good job in many aspects. The private dataset

they collected is highly convincing in proving their points. The

experimental design is also solid, with detailed results presented. The

methods are generally clearly described.

However, the major drawback that reduce the credibility of the

conclusion is that, only one object detection framework (YOLOX) was

analyzed in the manuscript. Although YOLOX is very successful in the

corresponding task, other frameworks such as faster RCNN, SSD, and

RetinaNe have also attained great success. The authors have to at least

provide some preliminary experimental evidence that the conclusions

obtained with YOLOX also apply to other frameworks.

Major problems include:

1.  Experiments have to be done with object detection networks other

    than YOLOX, so as to demonstrate the generalization of the

    conclusion. The authors may refer to this recent publication for

    candidate networks.

2.  The authors are encouraged to also include GAN methods in data

    augmentation.

3.  In Section 1., the authors introduce the application background of

    deep learning in medical image analysis. But the emphasis is on the

    overall development of convolutional neural network (CNN) rather

    than object detection networks. Please review the application of the

    deep learning in object detection.

4.  In Section 2.2., the authors mention that the values of CT data

    should be set within the 0-255 range to use for deep learning. But

    the HU value is always used in the preprocessing and normalization,

    which is rather confusing.

5.  The datasets of this study are from different hospitals and the

    chest CT protocols vary greatly. Is there standardization between

    different data sets in the preprocessing, such as getting rid of the

    CT bed? If not, whether the model performance is affected?

6.  In Section 2.5., the authors use FROC as an evaluation metric. But

    FROC is just a tool for characterizing the performance of a

    free-response system at all decision thresholds simultaneously. So

    is the the area under the empirical FROC curve (FAUC) actually used

    or something else?

The minor problems include:

1.  The full name of FROC is "free-response receiver operating

    characteristic" rather than "free receiver operating

    characteristic".

2.  The statements in the figures should be the same as those in the

    notes. For example, in Figure 4 on page 6, intersection over union

    uses the expression "IoU" in the picture, but "IOU" in the note.

3.  The author should pay more attention to the resolution of the

    figures, the axes and number of many figures are not clear.

4.  I suggest filling the columns with different patterns in the

    histogram, or at least improve the contrast of the colors.

5.  In Figure 2, the scales of the four datasets look different.

6.  In Section 2.6., the size of each set should be given.

7.  In Table 3, the representation of training set E (GNAH KNUH) is

    misleading.

8.  The authors should pay attention to the font size of figure

   annotations, as some of them are too small.

9.  The authors should pay attention to the use of italic vs roman font

    style in their equations.

Thank you!

Author Response

Dear Review3,

We would like to thank you and the reviewers of the Cancers for taking the time and effort to review our manuscript. Many of the valuable and constructive points that the reviewers raised, were appreciated by the authors. After considering the reviewers’ comments, we revised the manuscript and have indicated the corrections and changes made with yellow highlighting in the manuscript.

Comments to the Author

I am glad to review the manuscript entitled How Much Private Data Is Needed for Deep Learning in Lung Nodule Detection on CT scans? A Retrospective Multicenter Study. This manuscript provides guidelines for collecting lung nodule data to facilitate research. Although there are some major problems which have to be addressed through a significant revision, the overall quality of the research is satisfactory. 

In this manuscript, the authors set up five experimental configurations to solve three problems in the process of lung nodule data collection. Firstly, the authors compare the performance of a model trained on the LUNA16 dataset with that of a model trained on a private dataset. Then they confirm that pre-training on open data has a positive effect by comparing the results with and without transfer learning. Finally, the authors demonstrate the model performance with the different amount of private data.

 This manuscript did a very good job in many aspects. The private dataset they collected is highly convincing in proving their points. The experimental design is also solid, with detailed results presented. The methods are generally clearly described. However, the major drawback that reduce the credibility of the conclusion is that, only one object detection framework (YOLOX) was analyzed in the manuscript. Although YOLOX is very successful in the corresponding task, other frameworks such as faster RCNN, SSD, and RetinaNe have also attained great success. The authors have to at least provide some preliminary experimental evidence that the conclusions obtained with YOLOX also apply to other frameworks.

Major problems include:

  1. Experiments have to be done with object detection networks other than YOLOX, so as to demonstrate the generalization of the conclusion. The authors may refer to this recent publication for candidate networks.

Reply 1: Appreciate the constructive suggestion. We implemented one of the SOTA detection algorithm, EfficeintDet [1], and the results are shown in Figure 8, and Figure B. Also, the following description is added.

“To confirm generalization of our finding, we also implement another detection algorithm EfficientDet, and consistent results have been driven.”

[1] Tan, Mingxing, Ruoming Pang, and Quoc V. Le. "Efficientdet: Scalable and efficient object detection." Proceedings of the IEEE/CVF conference on computer vision and pattern recognition. 2020.

  1. The authors are encouraged to also include GAN methods in data augmentation.

Reply 2: We appreciate for the constructive suggestion. We agree that the reviewer's recommended GAN-based augmentation is a constructive approach to apply to our problem. However, the GAN-based augmentation method assumes that the GAN network should be well trained. There are many cases in which training of GAN is not successful and thus does not perform well compared to other augmentation techniques [2]. In addition, since our study focuses on confirming the performance of the detection model according to the number of training data, we mainly applied the augmentation method that does not require additional learning. However, we agree that the reviewer's suggestion can increase the ultimate detection performance, so we set it as our future research direction and mentioned it as follows in the discussion section.

“For the further study, we will focus on enhancing detection performance itself by adapting state-of-art techniques such generative adversarial networks-based augmentation.”

[2] Shorten, Connor, and Taghi M. Khoshgoftaar. "A survey on image data augmentation for deep learning." Journal of big data 6.1 (2019): 1-48.

  1. In Section 1., the authors introduce the application background of deep learning in medical image analysis. But the emphasis is on the overall development of convolutional neural network (CNN) rather than object detection networks. Please review the application of the deep learning in object detection.

Reply 3: We greatly appreciate the Reviewer’s comments. We agree completely and have revised as followed in Section 1:

“In this study, we used CNN-based object detection to detect lung nodules by analyzing CT scan images.”

  1. In Section 2.2., the authors mention that the values of CT data should be set within the 0-255 range to use for deep learning. But the HU value is always used in the preprocessing and normalization, which is rather confusing.

Reply 4: We appreciate the precise review. We agree completely and have revised as followed in Section 2.2:

"To use CT data for deep learning, we thus need to set their values within the 0–1 range."

  1. The datasets of this study are from different hospitals and the chest CT protocols vary greatly. Is there standardization between different data sets in the preprocessing, such as getting rid of the CT bed? If not, whether the model performance is affected?

Reply 5: We greatly appreciate the Reviewer’s helpful suggestion and agree with the Reviewer’s comment. As shown in Table 1, kVp/mA, Kernel, and slice/gap, which are chest CT protocols of Gangneung Asan Hospital (GNAH), Kangwon National University Hospital (KNUH), and Hallym Sacred Heart Hospital (HSHH), are not all the same, but very similar. This is because there should be no significant difference in the ability to detect abnormal lesions no matter which hospital a patient undergoes examination. This is also the reason why a similar Chest CT protocol is used in the medical field.

All Chest CTs of three institutions in this study are products of the same company from SIEMENS/SOMATOM, all consistent with a kVp of 120. Chest CT protocol of GNAH and KNUH is the same at B41f series kernel. In addition, the chest CT protocol of KNUH and HSHH is the same at 3 mm in slice/gap. In the preprocessing of the three organ datasets of GNAH, KNUH, and HSHH, standardization of getting rid of the CT bed was not performed, but the process of deep learning analysis by normalizing Hounsfield units, width center, and window width items of chest CT Dicom scan standardized image shape size and contrast.

In this research model that detects lung nodules, since it has already been learned that the chest wall structures such as muscles, bones, and clothes worn by the patient are not the pulmonary nodules, getting rid of the CT bed does not have a great effect on the model performance. And, since there is no mention of preprocessing to remove the CT bed in previous studies, it is thought that there are few things to think about in relation to the model performance.

  1. In Section 2.5., the authors use FROC as an evaluation metric. But FROC is just a tool for characterizing the performance of a free-response system at all decision thresholds simultaneously. So is the area under the empirical FROC curve (FAUC) actually used or something else?

Reply 6: We appreciate the precise review. We agree completely and have revised as followed in Section 2.5:

"Finally, based on this information, lung nodule detection performance was evaluated using the average precision (AP) and competition performance metric (CPM) utilizing free-response receiver operating characteristic (FROC). AP and CPM were used as quantitative evaluation metrics."

"FROC analysis is utilized for nodule detection evaluation in the LUNA16 challenge. In the FROC curve, sensitivity is plotted as a function of the average number of false positives per scan (FPs/scan). The CPM score is the average sensitivity at seven predefined false-positive rates (1/8, 1/4, 1/2, 1, 2, 4, and 8FPs per scan)."

In addition, we have revised the FROC metric to the CPM metric in experiments 1, 2, and 3.

The minor problems include:

  1. The full name of FROC is "free-response receiver operating characteristic" rather than "free receiver operating characteristic".

Reply 1: We apologize for this typo. We have corrected as followed in the Evaluation metrics:

"Finally, based on this information, lung nodule detection performance was evaluated using the average precision (AP) and competition performance metric (CPM) utilizing free-response receiver operating characteristic (FROC). AP and CPM were used as quantitative evaluation metrics. "

  1. The statements in the figures should be the same as those in the notes. For example, in Figure 4 on page 6, intersection over union uses the expression "IoU" in the picture, but "IOU" in the note.

Reply 2: We appreciate the precise review. We agree review’s comment and have revised figure 4 ("IoU" to "IOU").

  1. The author should pay more attention to the resolution of the figures, the axes and number of many figures are not clear.

Reply 3: We have enlarged axes and numbers in figures 7, 8, B1, and B2. Furthermore, we have submitted all the figures with vector graphics (.pdf) to ensure high resolution.

  1. I suggest filling the columns with different patterns in the histogram, or at least improve the contrast of the colors.

Reply 4: We appreciate the Reviewer’s helpful suggestion. We agree with the Reviewer’s comment and have revised figure 7 and figure A1.

  1. In Figure 2, the scales of the four datasets look different.

Reply 5: It is our intention. The amount of data acquired was different because the time when we start to acquire was different. So, we split the data based on time with ensuring the ratio of train, valid, and test set.

  1. In Section 2.6., the size of each set should be given.

Reply 6: We greatly appreciate the Reviewer’s comments. We agree with the Reviewer’s comment and have given the size of each dataset. The size of the existing private dataset is given in Section 2.1.4. Therefore, we additionally provided the subset size of LUNA16, which was not provided.

“The LUNA16 dataset consisted of 10 subsets. We used eight subsets (712 patients) for the training dataset and two subsets (176 patients) for the validation dataset. The private dataset included a total of 515 CT scans consisting of 181 data from KNUH, 287 data from GNAH, and 47 data from HSHH.”

  1. In Table 3, the representation of training set E (GNAH KNUH) is misleading.

Reply 7: We greatly appreciate the Reviewer’s comments. We have changed "GNAH KNUH" to "GNAH  KNUH".

  1. The authors should pay attention to the font size of figure annotations, as some of them are too small.

Reply 8: We appreciate the Reviewer’s helpful suggestion. We agree with the Reviewer’s comment and have revised all the figure annotations.

  1. The authors should pay attention to the use of italic vs roman font style in their equations.

Reply 7: Appreciate the detailed review. We reformulated all the equations. Please refer to the attached file.

Sincerely,
The authors.

Round 2

Reviewer 3 Report

The authors have gone into great lengths to revise the manuscript, and I am happy to witness the great improvement of the manuscript. I appreciate the authors for their rigorousness.

However, there is still something which could be further enhanced. In Major Point 3 of the previous round of revision, I suggested the authors to "review the application of the deep learning in object detection". This may include a concise introduction of some SOTA implementations of CNNs in the task of object detection. The authors may mention YOLO, EffifcientDet, and other classical networks in this part.

However, the mere mentioning of the connections between CNNs and the object detection task is not sufficient. I understand that this manuscript mainly focuses on data, nevertheless, the model itself is worthy of being mentioned in 2-3 sentences.

Finally, the authors did a great work in improving the figures and equations, however, there are still some parts that are not very suitable in typesetting. This part might be left to the discretion of the typesetters.

Author Response

Dear Review3,

We would like to thank you and the reviewers of the Cancers for taking the time and effort to review our manuscript. Many of the valuable and constructive points that the reviewers raised, were appreciated by the authors. After considering the reviewers’ comments, we revised the manuscript and have indicated the corrections and changes made with yellow highlighting in the manuscript.

Comments to the Author

 The authors have gone into great lengths to revise the manuscript, and I am happy to witness the great improvement of the manuscript. I appreciate the authors for their rigorousness.

However, there is still something which could be further enhanced. In Major Point 3 of the previous round of revision, I suggested the authors to "review the application of the deep learning in object detection". This may include a concise introduction of some SOTA implementations of CNNs in the task of object detection. The authors may mention YOLO, EffifcientDet, and other classical networks in this part.

  1. However, the mere mentioning of the connections between CNNs and the object detection task is not sufficient. I understand that this manuscript mainly focuses on data, nevertheless, the model itself is worthy of being mentioned in 2-3 sentences.

Reply 1: We appreciate the Reviewer’s helpful suggestion. We agree with the Reviewer’s comment and have revised as followed in Section 1:

“CNN-based object detection tasks are making a lot of progress. Faster R-CNN [43], RetinaNet [44], EfficientDet [42], and YOLOX [32], etc. which were state-of-art models at the time, show the development of object detection. In this study, we experimented with lung nodule detection based on the YOLOX model. The specific details of the model used are described in Section 2.4.”

[32]       Ge, Z.; Liu, S.; Wang, F.; Li, Z.; Sun, J. Yolox: Exceeding yolo series in 2021. arXiv preprint arXiv:2107.08430 2021.

[42]       Tan, M.; Pang, R.; Le, Q. V. Efficientdet: Scalable and efficient object detection. In Proceedings of the IEEE/CVF conference on computer vision and pattern recognition, Seattle, WA, USA, 13-19 June 2020; pp. 10781-10790.

[43]       Ren, S.; He, K.; Girshick, R.; Sun, J. Faster R-CNN: Towards real-time object detection with region proposal networks. In Proceedings of the International Conference on Neural Information Processing Systems, Montreal, QC, Canada, 7–12 December 2015; pp. 91–99.

[44]       Lin, T.-Y.; Goyal, P.; Girshick, R.; He, K.; Dollár, P. Focal loss for dense object detection. In Proceedings of the Proceedings of the IEEE international conference on computer vision, Venice, Italy, 22–29 October 2017; pp. 2980-2988.

  1. Finally, the authors did a great work in improving the figures and equations, however, there are still some parts that are not very suitable in typesetting. This part might be left to the discretion of the typesetters.

Reply 2: We appreciate for the detailed review. We agree with the reviewer's comment. To check the suitability in typesetting, we have printed all the figure in greyscale, and have revised the figure 6, 8, B2, and B3.

Sincerely,
The authors.
